# Cardiorespiratory Fitness without Exercise Testing Can Predict All-Cause Mortality Risk in a Representative Sample of Korean Older Adults

**DOI:** 10.3390/ijerph16091633

**Published:** 2019-05-10

**Authors:** Moongu Song, Inhwan Lee, Hyunsik Kang

**Affiliations:** College of Sport Science, Sungkyunkwan University, Suwon 16419, Korea; somogo@naver.com (M.S.); ansh00@skku.edu (I.L.)

**Keywords:** aging, gender, mortality, physical fitness

## Abstract

This study examined the association between cardiorespiratory fitness (CRF) without exercise testing and all-cause mortality in Korean older adults. The present study was carried out using data from the 2008 and 2011 Living Profiles of Older People Survey. A total of 14,122 participants aged 60 years and older (57% women) completed the 2008 baseline and 2011 follow-up assessments (i.e., socioeconomic status, health behaviors and conditions, and prevalence of chronic diseases), and they were included for the final analyses. CRF was estimated (eCRF) with sex-specific algorithms and classified as lower (lowest 25%), middle (middle 50%), and upper (highest 25%). Cox proportional hazards regression was used to calculate hazard ratios (HRs) and 95% confidence intervals (CIs) across eCRF categories. In total, multivariable-adjusted HRs and 95% CIs were 1 for the upper eCRF group (referent), 1.059 (0.814~1.378) for the middle eCRF group, and 1.714 (1.304~2.253) for the lower eCRF group. In men, multivariable-adjusted HRs and 95% CIs were 1 for the upper eCRF group (referent), 1.011 (0.716~1.427) for the middle eCRF group, and 1.566 (1.098~2.234) for the lower eCRF group. In women, multivariable-adjusted HRs and 95% CIs were 1 for the upper eCRF group (referent), 1.064 (0.707~1.602) for the middle eCRF group, and 1.599 (1.032~2.478) for the lower eCRF group. The current findings suggest that eCRF may have an independent predictor of all-cause mortality, underscoring the importance of promoting physical activity to maintain a healthful level of CRF in Korean geriatric population.

## 1. Introduction

Cardiorespiratory fitness (CRF) is a key component of health-related physical fitness and reflects the ability of the respiratory and circulatory systems to deliver oxygen to working skeletal muscles during exercise. Maintenance and enhancement of CRF influence several biologic mechanisms in such way that the risk of various chronic diseases may be reduced significantly [1,2]. In addition, CRF was found to alter the relationship between adiposity and its health outcomes, as illustrated in the so-called the obesity paradox [3]. In this context, CRF is also known as a strong predictor of all-cause and cardiovascular disease mortality [4], cancer mortality [5], as well as incidence of various chronic diseases [6], supporting its diagnostic and prognostic values in the prediction of premature death from all causes and specific causes.

While superior to estimated CRF (eCRF), objectively measuring CRF is often limited by the need of specialized equipment and trained personnel and time-consuming [7] as well as mobility problems and chronic conditions in elderly persons [8]. However, the findings of recent studies showed that CRF could be estimated from easily obtained health indicators with an acceptable accuracy [9]. Thus, CRF without exercise testing may provide an alternative approach for routine clinical screening and health promotion with the purpose to identify individuals with poor CRF who are at increased health risk [10,11]. 

Although mostly limited to Western populations, some studies showed that eCRF was inversely associated with mortality from all causes and specific causes. For example, the outcomes of the 1999–2006 National Health and Nutrition Examination Survey [10] and the Aerobics Center Longitudinal Study [11] showed that eCRF was significantly and inversely associated with all-cause and CVD mortality. In a Spanish cohort of men and women aged 60 years and older, however, Martinez-Gomez et al. [12] found that eCRF was inversely associated with all-cause mortality in women but not in men, questioning the value of non-exercise eCRF as a diagnostic and/or prognostic tool to predict mortality in elderly adults. 

Declines in the physical ability to tolerate daily activities may predict mobility problems and mortality from all causes and specific causes, particularly in the sedentary older populations [13,14]. Thus, maintaining a healthful level of CRF via physical activity is of paramount importance in attenuating morbidity and mortality from various chronic diseases and thereby reducing their societal and economic burden. Yet, little information is available regarding the clinical values of eCRF in predicting mortality of Asian populations, including Koreans. Therefore, this study examined the association between eCRF and all-cause mortality during a 3-year follow-up in a representative sample of Korean adults aged 60 years and older.

## 2. Materials and Methods

### 2.1. Study Design and Participants

The present study was carried out using data from the 2008 and 2011 Living Profiles of Older People Survey (LPOPS). The designs and methods of the LPOPS have been described previously [15]. In brief, the survey used stratified two-stage cluster sampling. The primary sampling unit was based on the 2005 census frame, with secondary sampling units consisting of households with older residents. The strata consisted of 7 metropolitan and 18 provincial (urban and rural) regions. Sampling within secondary geographic strata used auxiliary stratification indices, such as sex ratio and average age obtained from surveys, to yield a representative sample.

The baseline survey, wave 1, was carried out in 2008, and a follow-up survey; wave 2, was carried out in 2011. In wave 1, a total of 19,007 non-institutionalized and community-dwelling adults aged ≥60 years were initially invited to the Living Profiles of Older People Survey. Among the invited older adults, 15,146 agreed to participate in the 2008 baseline assessment, and 3861 were excluded due to refusal, visual impairment, hearing impairment, and language impairment (response rate 79.5%). In wave 2, 1024 individuals were additionally excluded due to no data to estimate CRF, refusal, hospitalization, institutionalization, or loss of contact. Consequently, a total of 14,122 participants (57% women) completed the 2011 follow-up assessments, and they were included for the final analyses. The institutional review board of human study reviewed and approved the study protocol participants (SKKU 2017-06-009). Informed consent was obtained from all of the study. 

### 2.2. Study Variables

#### 2.2.1. Determination of CRF without Exercise Testing

CRF without exercise testing at baseline was estimated in peak minute volume of oxygen consumption (VO_2peak_) in the unit of mL·min^−1^·kg^−1^ with the following sex-specific algorithms [16]:
Estimated CRF (women): 56.363 + 1.921 (Physical Activity Rating) − 0.381 (Age) − 0.754 (Body Mass Index) and 
Estimated CRF (men): 67.350 + 1.921 (Physical Activity Rating) − 0.381 (Age) − 0.754 (Body Mass Index).

Physical activity status of study participants over the preceding month was assessed with using the Johnson Space Centre (JSC) Physical Activity Rating (PAR) scale. The JSC-PAR scale has been developed to provide a self-rating in the range of 0 (avoid physical activities whenever possible) to 7 (heavy physical activities done regularly for more than 3 h per week) [17]. Participants are to select only one response that best describes their physical activity level. The participants were asked to select the number (0 to 7) that best described their general level of physical activity for the previous month. 

In a previous study involving both men and women aged 20~79 years, the JSC physical activity scale was found to have a strong and independent relationship with maximal oxygen uptake (VO_2peak_) measured with a maximal walking treadmill test [9]. A generalized regression model for VO_2peak_ estimation had correlation coefficient of 0.88 (standard error of estimate, SEE = 4.90 mL·min^−1^·kg^−1^). Gender-specific regression models for VO_2peak_ estimation had 0.85 (SEE = 4.64 mL·min^−1^·kg^−1^) and 0.85 (SEE = 5.02 mL·min^−1^·kg^−1^) for women and men, respectively [9].

Once the algorithms were implemented, participants were classified into lower (lowest 25%), middle (middle 50%), and upper (highest 25%) on the base of sex-specific tertiles of the estimated VO_2peak_ distribution.

#### 2.2.2. Determination of All-Cause Mortality

All-cause mortality was defined as number of deaths from all causes, which was identified by death certificate filed upon at the registry office of the municipality of residence. Follow-up time was defined as the period from the baseline visit to the day of death for participants who died or the last contact date (12 August 2011) for those who did not have the outcome event (censored).

#### 2.2.3. Determination of Covariates

The study measured social and demographic factors including age, gender, height, weight, education, and monthly income. Health behavioral factors were measured and included smoking, alcohol consumption, solitary, and nutritional status, as described in detail elsewhere [15,18]. Smoking status was categorized as past/current smoker or non-smoker. Alcohol consumption was classified as abstinent, moderate (1–3 drinks/week), or heavy (more than the moderate level of consumption). Nutritional status was assessed using the nutrition screening initiative checklist and classified as low (0–2), moderate (3–5), or high risk (≥ 6) according to the checklist score, as described in detail elsewhere [19].

Measured parameters regarding health conditions included frailty, impaired activities of daily living (ADL), experiences of falling and hospitalization, impaired cognition, and depressive symptoms. A modified version of the original CHS frailty index was used to categorize frailty as robust, pre-frail, or frail, as described in detail elsewhere [20]. Cognitive impairment was defined as scoring more than 1.5 standard deviations below the age-, gender-, and education-adjusted norms of the Mini-Mental State Examination, as described in detail elsewhere [21]. Depressive symptoms was defined with a score of ≥8 on the 15-item, short-form of the self-administered Korean version of Geriatric Depression Scale, as described in detail elsewhere [22]. Physician-diagnosed chronic conditions, including hypertension, diabetes, coronary artery diseases (CAD), stroke, and lung diseases were also included. 

Height and body weight were measured using a portable digital scale (JENIX^®^, Dong Shan Jenix Co., TLD., Seoul, Korea), after removing the shoes and wearing only light clothing. Body mass index (BMI) was calculated by dividing body weight by height squared (kg/m^2^). 

### 2.3. Statistical Analyses

All variables were checked for normality, both visually and through the Kolmogorov–Smirnov test, and subjected to log^10^ transformation, if necessary, prior to statistical analyses. Multiple imputations by chained equation (MICE) was performed to handle the missing data of the study variables identified (<26%) prior to the primary analyses. The number of imputations was determined to be four.

Descriptive statistics were presented as means and standard deviations and frequencies and percentages for continuous and categorical variables, respectively. Linear regressions and Chi-square were used to compare differences in means and percentages of measured variables, respectively. The Kaplan–Meier procedure with log-rank tests was used to estimate mortality functions according to baseline eCRF categories. Survival time was measured as the time from the baseline survey to death or the censor point (12 August 2011). Cox proportional hazards regression model was used to estimate the hazard ratio (HR) and 95% confidence interval (95% CI) of baseline eCRF categories for 3-year all-cause mortality as censored observation before and after adjustments for all measured covariates (i.e., socio-demographic variables, health behaviors and conditions, and chronic conditions). To minimize the possibility of a reverse causation between eCRF and mortality risk due to a relatively short follow-up period, a sensitivity analysis was additionally performed by excluding all the events that occurred during the first year of the 3-year follow-up period. All analyses were performed taking into account complex sampling weights, using SPSS-PC 23.0 (SPSS Inc., Chicago, IL, USA).

## 3. Results

Table 1 describes the associations between eCRF and baseline characteristics in this study population. In total, a large portion of the subjects in the higher eCRF groups were younger age; had lower BMI; and had higher PAR and estimated VO_2_. With respect to health behaviors, eCRF was positively associated with education, income, and alcohol consumption but inversely with solitary and nutritional risk. With respect to health conditions, a large portion of the subjects in the high eCRF groups had lower percentages of frailty, impaired ADL, hospitalization, falling, impaired cognition, and depressive symptoms. Additionally, a large portion of subjects in the higher eCRF groups had lower prevalence of hypertension, diabetes, CAD, stroke, and lung diseases. No association between eCRF and smoking was found.

In both men and women, a large portion of the subjects in the higher eCRF groups were younger, had lower BMI, and had higher PAR and estimated VO_2_. With respect to health behaviors in both men and women, a large portion of the subjects in the higher eCRF groups were more educated, less solitary, had higher income and smoking, and had lower nutritional risk. With respect to health conditions in both men and women, a large portion of the subjects in the higher eCRF groups had lower percentages of frailty, impaired ADL, hospitalization, falling, and depressive symptom. Such association between eCRF and impaired cognition was observed. In addition, a large portion of the subjects in the higher eCRF groups had lower prevalence of hypertension, diabetes, CAD, stroke, and lung diseases in both men and women. However, an association between eCRF and alcohol consumption was found in men only but not in women.

During an average follow-up of 3 years, 440 men and 261 women (4.9%) with baseline measurements died from all causes. Figure 1 illustrates survival curves during the 3-year follow-up period according to eCRF categories. In total, the mortality rate per 1000 person-years was 3.0 for the lower eCRF group, 2.9 for the middle eCRF group, and 1.1 for the upper eCRF group. The Kaplan–Meier mortality functions, as illustrated in Figure 1A, showed that the upper and middle eCRF groups had a significantly higher risk of all-cause mortality (*p* < 0.001 and *p* < 0.001, respectively) compared with the lower eCRF group, with no difference in all-cause mortality between the upper and middle eCRF groups (*p* = 0.073). 

In men, the mortality rate per 1000 person-years was 1.2 for the lower eCRF group, 0.6 for the middle eCRF group, and 0.5 for the upper eCRF group. The Kaplan–Meier mortality functions, as illustrated in Figure 1B, showed that the upper and middle eCRF groups had a significantly higher risk of all-cause mortality (*p* < 0.001 and *p* < 0.001, respectively) compared with the lower eCRF group, with no difference in all-cause mortality between the upper and middle eCRF groups (*p* = 0.128). 

In women, the mortality rate per 1000 person-years was 0.6 for the lower eCRF group, 0.3 for the middle eCRF group, and 0.2 for the upper eCRF group. The Kaplan–Meier morality functions, as illustrated in Figure 1C, showed that the upper and middle eCRF groups had a significantly lower risk of all-cause mortality (*p* < 0.001 and *p* < 0.001, respectively) compared with the lower eCRF group, with no difference in all-cause mortality between the upper and middle eCRF groups (*p* = 0.066).

Table 2 represents the associations between eCRF and all-cause mortality. In total, trend testing showed significant and linear trends in both 3- and 2-year all-cause mortality across incremental eCRF categories (*p* < 0.001 and *p* < 0.001, respectively); considering the upper eCRF group as reference (HR = 1), the lower and middle CRF groups had a higher 3-year all-cause mortality risk (HR = 2.648 and 95% CI = 2.121~3.307; HR = 1.313 and 95% CI = 1.048~1.645, respectively) as well as a 2-year all-cause mortality risk (HR = 2.810 and 95% CI = 2.156~3.663; HR = 1.445 and 95% CI = 1.106~1.889, respectively). The linear trend for crude HRs remained statistically significant even after adjustments for socioeconomic status, health behaviors and conditions, and prevalence of chronic diseases. 

In men, trend testing showed significant and linear trends in both 3- and 2-year all-cause mortality across incremental eCRF categories (*p* < 0.001 and *p* < 0.001, respectively); considering the upper eCRF group as reference (HR = 1), the lower and middle CRF groups had a higher 3-year all-cause mortality risk (HR = 2.227 and 95% CI = 1.673~2.967; HR = 1.253 and 95% CI = 0.936~1.676, respectively) as well as a 2-year all-cause mortality risk (HR = 2.252 and 95% CI = 1.601~3.168; HR = 1.366 and 95% CI = 0.969~1.926, respectively). The liner trend for crude HRs remained statistically significant even after adjustments for all the covariates.

In women, trend testing showed significant and linear trends in 3- and 2-year all-cause mortalities across incremental eCRF categories (*p* < 0.001 and *p* < 0.001, respectively); considering the upper eCRF group as reference (HR = 1), the lower and middle eCRF groups had a higher 3-year all-cause mortality risk (HR = 3.056 and 95% CI = 2.151~4.343; HR = 1.393 and 95% CI = 0.976~1.989) as well as a 2-year all-cause mortality risk (HR = 3.479 and 95% CI = 2.281~5.306; HR = 1.558 and 95% CI = 1.106~2.390, respectively). The linear trend for crude HRs remained statistically significant, even after adjustments for all the covariates.

## 4. Discussion

In this population-based prospective study, we found that a higher eCRF was generally associated with more favorable profiles of socioeconomic status, health behaviors, and conditions, as well as less prevalence of chronic diseases. To the best of our knowledge, this is the first study documenting a strong predictive value of eCRF for all-cause mortality during an average follow-up of 3 years in Korean older adults even after adjustments for all the covariates assessed at baseline. 

The current findings support and extend those of previous studies reporting an inverse association between eCRF and mortality from all causes and diseases-specific causes. By analyzing an longitudinal survey data from the 1988–2011 NHANES in which 12,834 non-Hispanic white and black and Mexican American aged 20–86 years were followed up for 19.2 years, Zhang et al. [23] showed that after adjustments for race/ethnicity, education, age, hypertension, diabetes, hypercholesterolemia, CVD, and cancer status, men in the middle and upper eCRF groups had a significantly lower mortality from all causes and CVD compared with men in the lower eCRF group. Likewise, women in the middle and upper eCRF groups had a significantly lower mortality from all causes and CVD even after adjustments for several confounders compared with women in the lower eCRF group. 

In a pooled sample of population-based cohorts involving 32,319 adults aged 35–70 years who completed baseline examinations of the 1994, 1998, 1999, 2004 Health Survey for England and the 1995 Scottish Health Survey, Stamatakis et al. [24] showed that a higher eCRF score was significantly associated with a lower risk of mortality from all-causes during a mean follow-up of 9 years and CVD even after adjustments for diabetes, hypertension, smoking, social class, alcohol, and depression.

In a subsample of 8936 adults aged 35–70 years without baseline CVD among those who participated in the 1991–1994 Copenhagen City Heart Study, Holtermann et al. [25] showed that self-reported CRF was inversely associated with CVD and all-cause mortality during an average follow-up of 17.9 years in both men and women. Furthermore, they showed that adding self-reported CRF to traditional risk factors resulted in net reclassification improvements of 30.5% and 25.4% for CVD and all-cause mortality, respectively. 

On the other hand, two previous studies suggested that gender might play as a potential confounder in determining the association between eCRF and all-cause and/or cause-specific mortality, especially in geriatric populations. By analyzing data obtained from 37,112 healthy Norwegians aged 20 years and older who participated in the first HUNT study, Nes et al. [26] found that eCRF based on algorithms developed by Nes et al. [27] was inversely associated with all-cause and CVD mortality during a mean follow-up of 24 year in both men and women below 60 years of age even after adjustments for potential confounders, including age, smoking, alcohol consumption, marital status, family history of disease, and education. However, a separate analysis using those aged 60 years and older only showed that an inverse association between eCRF and all-cause and CVD-mortality was found in women but not in men. 

In a representative Spanish sample consisting of non-institutionalized 1739 men and 2269 women aged 60 years and older, Martinez-Gomez et al. [12] examined whether or not eCRF based on the sex-specific algorithms developed by Jackson et al. [28] was associated with all-cause mortality during an average follow-up of 9.4 years. They found that compared with women in the lowest quartile of CRF, the HR (95% CI) for all-cause death was 0.81 (0.62~1.06) in the second quartile, 0.68 (0.48~0.95) in the third quartile, and 0.56 (0.36~0.87) in the highest quartile (*p* for linear trend = 0.004). Unlike women, however, no association was found between eCRF and all-cause mortality among men.

Unlike the previous studies reporting a gender-dependent association between eCRF and all-cause mortality, we found that regardless of gender, eCRF based on algorithms developed by Jackson et al. [18] was inversely associated with all-cause mortality in Korean older adults aged 60 years and older even after adjustments for socioeconomic status, health behaviors and conditions, and prevalence of chronic diseases. The sample size was relatively large, an adequate number of deaths were observed even during a relatively short follow-up of 3 years, and a number of covariates, including socioeconomic status, health behaviors, health conditions, and prevalence of various chronic diseases, were included as potential confounders in the risk prediction of eCRF for premature death from all causes. In addition, the outcomes of the sensitivity analysis (i.e., including all the events occurred during the last two years) were similar to those of the primary analysis (i.e., including all the events occurred during the entire 3-year follow-up), confirming an inverse association between eCRF and all-cause mortality.

Differences in follow-up period, race/ethnicity, and algorithms used to estimate CRF, and covariates might be part of the reasons why two previous studies found a gender-specific association between eCRF and all-cause mortality or others did not. A well-designed longitudinal study will be necessary to confirm whether or not gender plays a role in determining the association between eCRF and mortality from all causes and diseases in the geriatric populations.

Several explanations can be given for the inverse association between eCRF and all-cause mortality observed in the current study. The protective effect of eCRF against premature death from all causes might be achieved by adopting healthy lifestyles featuring of healthy diets and physical activity, which, in turn, reduce metabolic complications including elevated blood pressure [29] and fasting glucose [30], and atherogenic lipids [31], improve insulin sensitivity [32], vascular health [33], and immune function [34], control body weight [35], and enhance cardiorespiratory function [36] and cognitive function [37]. In a long-term perspective, those health benefits would decrease the risk of being exposed to various chronic diseases, as well as the risk of premature death from all causes and specific causes.

This study has also some limitations. First, the follow-up period of 3 years is relatively short, which may underestimate the values of eCRF as a prognostic and diagnostic tool in the prediction of mortality from all causes. Second, eCRF was obtained at baseline only, so it is possible that eCRF might change during follow-up because components used to estimated eCRF may change, which primarily would lead to an underestimation of the study association. Third, the non-exercise algorithm we applied was derived from a cross-sectional study of which eCRF was based on linear regression of age. However, recent studies showed that CRF declines nonlinearly with aging [38]. The algorithms derived from the linear regression of aging may provide a less precise estimation of CRF compared to the algorithms derived from the linear regression of aging [27]. 

## 5. Conclusions

Low CRF is well known as an independent indicator of premature death from all and specific causes. However, objectively measuring CRF is usually limited due to mobility issues, chronic conditions, and other health complications in elderly persons. In this context, therefore, we investigated the role of CRF without exercise testing in the prediction of all-cause mortality in a representative sample of Korean older adults and found that a higher eCRF was significantly associated with a lower risk of all-cause mortality, implying a predictive value of eCRF for all-cause mortality in Korean older adults. 

## Figures and Tables

**Figure 1 ijerph-16-01633-f001:**
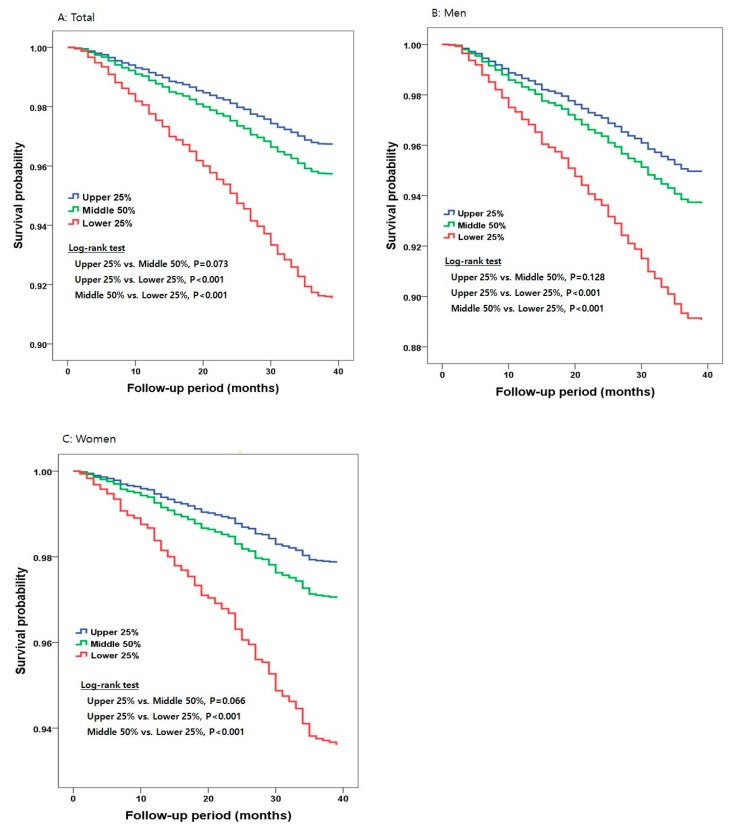
Survival curves according to eCRF categories in total (**A**), men (**B**), and women (**C**).

**Table 1 ijerph-16-01633-t001:** Descriptive statistics of baseline characteristics according to eCRF categories.

	eCRF											
	Total (*N* = 14,122, 100%)				Men (*N* = 6105, 43.2%)				Women (*N* = 8017, 56.8%)			
	Lower (*n* = 3529)	Middle (*n* = 7063)	Upper (*n* = 3530)	*p* for Linear Trends	Lower (*n* = 1525)	Middle (*n* = 3054)	Upper (*n* = 1526)	*p* for Linear Trends	Lower (*n* = 2004)	Middle (*n* = 4009)	Upper (*n* = 2004)	*p* for Linear Trends
Age (years)	76.2 ± 7.1	68.3 ± 5.6	65.2 ± 4.5	<0.001	74.8 ± 6.8	67.0 ± 5.0	65.4 ± 4.5	<0.001	77.2. ± 7.1	69.2 ± 5.9	64.9 ± 4.4	<0.001
BMI (kg/m^2^)	25.6 ± 3.5	23.5 ± 2.8	22.4 ± 2.9	<0.001	24.8 ± 3.1	22.9 ± 2.6	22.6 ± 3.0	<0.001	26.2 ± 3.7	24.0 ± 2.9	22.3 ± 2.9	<0.001
PAR (score)	0.77 ± 0.53	1.26 ± 0.87	3.88 ± 2.25	<0.001	0.88 ± 0.54	1.47 ± 1.01	5.09 ± 1.98	<0.001	0.69 ± 0.51	1.11 ± 0.72	2.97 ± 2.00	<0.001
Estimated VO_2_ (mL·min^−1^·kg^−1^)	14.3 ± 7.0	19.8 ± 6.8	26.8 ± 7.9	<0.001	21.8 ± 2.2	27.3 ± 1.8	35.2 ± 3.0	<0.001	8.5 ± 2.1	14.1 ± 1.6	20.5 ± 3.2	<0.001
Parameters of health behaviors												
Education (years)	5.0 ± 4.9	6.5 ± 4.7	7.9 ± 4.7	<0.001	7.6 ± 4.9	8.7 ± 4.4	9.6 ± 4.3	<0.001	3.0 ± 3.8	4.9 ± 4.2	6.6 ± 4.5	<0.001
Income (1000 won)	1410.9 ± 2151.8	1559.0 ± 1637.1	1846.5 ± 2058.3	<0.001	1541.7 ± 2207.3	1793.6 ± 1763.5	2142.5 ± 2475.1	<0.001	1311.4 ± 2103.6	1380.3 ± 1509.8	1621.2 ± 1637.9	<0.001
Solitary (%)	831 (23.6)	1118 (15.8)	375 (10.6)	<0.001	129 (8.5)	157 (5.1)	62 (4.1)	<0.001	702 (35.0)	961 (24.0)	313 (15.6)	<0.001
Nutritional risk (%)	2013 (57.1)	3085 (43.7)	1249 (35.4)	<0.001	778 (51.0)	1179 (38.6)	464 (30.4)	<0.001	1235 (61.6)	1905 (47.5)	785 (39.2)	<0.001
Smoking (%)	1227 (34.8)	2507 (35.5)	1200 (34.0)	0.489	1028 (67.4)	2276 (74.5)	1102 (72.2)	0.003	199 (9.9)	231 (5.8)	97 (4.8)	<0.001
Alcohol intake (%)	494 (14.0)	1265 (17.9)	669 (19.0)	<0.001	406 (26.6)	1090 (35.7)	561 (36.8)	<0.001	88 (4.4)	175 (4.4)	108 (5.4)	0.133
Parameters of health conditions												
Frailty (%)	340 (14.5)	287 (5.4)	45 (1.6)	<0.001	152 (14.1)	114 (4.8)	12 (0.9)	<0.001	188 (14.8)	174 (5.9)	33 (2.1)	<0.001
Impaired ADL (%)	482 (13.7)	443 (6.3)	111 (3.1)	<0.001	194 (12.7)	152 (5.0)	51 (3.3)	<0.001	289 (14.4)	290 (7.2)	60 (3.0)	<0.001
Hospitalization (%)	561 (15.9)	874 (12.4)	363 (10.3)	<0.001	232 (15.2)	377 (12.3)	118 (7.7)	<0.001	328 (16.4)	497 (12.4)	245 (12.2)	<0.001
Falling (%)	682 (19.3)	981 (13.9)	372 (10.5)	<0.001	203 (13.3)	296 (9.7)	106 (6.9)	<0.001	479 (23.9)	685 (17.1)	266 (13.3)	<0.001
Impaired cognition (%)	889 (26.5)	1440 (21.4)	646 (19.1)	<0.001	481 (33.2)	694 (23.8)	284 (19.5)	<0.001	408 (21.5)	745 (19.6)	362 (18.9)	0.044
Depressive symptoms (%)	1303 (37.3)	1715 (24.5)	520 (14.8)	<0.001	462 (30.8)	594 (19.6)	153 (10.1)	<0.001	841 (42.3)	1121 (28.2)	367 (18.5)	<0.001
Prevalence of chronic diseases												
Hypertension (%)	2017 (57.2)	2968 (42.0)	1255 (35.6)	<0.001	777 (51.0)	1137 (37.2)	485 (31.8)	<0.001	1241 (61.9)	1832 (45.7)	769 (38.4)	<0.001
Diabetes (%)	659 (18.7)	1099 (15.6)	415 (11.8)	<0.001	269 (17.6)	452 (14.8)	158 (10.4)	<0.001	390 (19.5)	647 (16.1)	257 (12.8)	<0.001
CAD (%)	347 (9.8)	489 (6.9)	211 (6.0)	<0.001	135 (8.9)	204 (6.7)	85 (5.6)	<0.001	212 (10.6)	284 (7.1)	126 (6.3)	<0.001
Stroke (%)	279 (7.9)	366 (5.2)	112 (3.2)	<0.001	173 (11.3)	187 (6.1)	35 (2.3)	<0.001	107 (5.3)	179 (4.5)	77 (3.8)	0.023
Lung diseases (%)	240 (6.8)	338 (4.8)	132 (3.7)	<0.001	128 (8.4)	167 (5.5)	76 (5.0)	<0.001	112 (5.6)	172 (4.3)	55 (2.7)	<0.001

mL·min^−1^·kg^−1^ is defined as milliliter (mL) of oxygen consumption per minute (min) per body weight (kg). eCRF: estimated cardiorespiratory fitness. PAR: Physical Activity Rating. ADL: Activities of Daily Living.

**Table 2 ijerph-16-01633-t002:** Risks of all-cause mortality according to eCRF category (HR and 95% CI).

	3-Year Mortality Risk		2-Year Mortality Risk	
	Model 1	Model 2	Model 1	Model 2
Total (*N* = 14,122)				
Upper	1 (reference)	1 (reference)	1 (reference)	1 (reference)
Middle	1.313 (1.048–1.645)	1.059 (0.814–1.378)	1.445 (1.106–1.889)	1.189 (0.868–1.628)
Lower	2.648 (2.121–3.307)	1.714 (1.304–2.253)	2.810 (2.156–3.663)	1.883 (1.358–2.612)
*p* for linear trends	<0.001	<0.001	<0.001	<0.001
Men (*N* = 6105)				
Upper	1 (reference)	1 (reference)	1 (reference)	1 (reference)
Middle	1.253 (0.936–1.676)	1.011 (0.716–1.427)	1.366 (0.969–1.926)	1.084 (0.724–1.623)
Lower	2.227 (1.673–2.967)	1.566 (1.098–2.234)	2.252 (1.601–3.168)	1.653 (1.090–2.508)
*p* for linear trends	<0.001	<0.001	<0.001	<0.001
Women (*N* = 8017)				
Upper	1 (reference)	1 (reference)	1 (reference)	1 (reference)
Middle	1.393 (0.976–1.989)	1.064 (0.707–1.602)	1.558 (1.106–2.390)	1.274 (0.769–2.112)
Lower	3.056 (2.151–4.343)	1.599 (1.032–2.478)	3.479 (2.281–5.306)	1.865 (1.089–3.195)
*p* for linear trends	<0.001	<0.001	<0.001	<0.001

eCRF: estimated cardiorespiratory fitness; HR: hazard ratio; CI = confidence interval. Model 1: Unadjusted; Model 2: Adjusted for socioeconomic status (i.e., education and income), socioeconomic status and health behaviors (i.e., solitary, nutritional risk, smoking, alcohol consumption) and conditions (i.e., frailty, impaired ADL, hospitalization, falling, impaired cognition, and depressive symptoms), and chronic diseases (i.e., hypertension, diabetes, CAD, stroke, lung disease, and cancer).

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
