# Peer review of "Cardiorespiratory Fitness without Exercise Testing Can Predict All-Cause Mortality Risk in a Representative Sample of Korean Older Adults"

_ijerph, 2019, doi:10.3390/ijerph16091633_

Round 1

Reviewer 1 Report

The principal aim of this study was to investigate association between non exercice eCRF and all- cause mortality in Korean older adults. The results showed that regardless of gender, non-exercise eCRF was inversely associated with all-cause mortality during an average follow-up of 3 years in Korean older adults aged 60 years and older even after adjustments for socioeconomic status, health behaviors and conditions, and prevalence of chronic diseases. The authors suggested that eCRF may have a strong predictive value for all-cause mortality in Korean geriatric population. The study is well-designed and well-conducted, and the results are interesting and substantial for the community. However, authors should provide some important information to refine their article.

Comments:

Study variables

- Lines 85-86: In this section the authors say that the JSC physical activity scale has a strong independent relationship with maximal oxygen uptake in men and women between the ages of 20 and 79 years: In my opinion, the authors should specify the correlation coefficient and the test used to measure the maximum consumption among the participants of reference [11]

Statistical analyses

- Line 117: The authors used linear regressions to compare differences in means. The authors did not specify whether they verified the premise of using linear regression (normal distribution, Homogeneity of variances, Homoscedasticity). I therefore suggest that the authors give these details in their paper.

Results

- Lines 134-135: How the authors explain the lack of association between eCRF and cancers in men.

- I suggest to the authors to enlarge Figure 1

Discussion

- Line 241: I think it's complications instead of complicstions.

- Lines 238-244: The authors need to delve deeper into each possible explanation of the correlation between eCRF and all-cause mortality. For example how adopting healthy lifestyles featuring of healthy diets and physical activity enhance cognitive function?

Author Response

(Reviewer #1)

Q#1) Lines 85-86: In this section the authors say that the JSC physical activity scale has a strong independent relationship with maximal oxygen uptake in men and women between the ages of 20 and 79 years: In my opinion, the authors should specify the correlation coefficient and the test used to measure the maximum consumption among the participants of reference [11]

ANS#1) Thanks for the comments. In response to the comments, the following statements are now revised and added.

“In a previous study involving both men and women aged 20~79 years, the JSC physical activity scale was found to have a strong and independent relationship with maximal oxygen uptake (VO2peak) measured with a maximal walking treadmill test [9]. A generalized regression model for VO2peak estimation had correlation coefficient of 0.88 (SEE=4.90 ml× min-1×kg-1). Gender-specific regression models for VO2peak estimation had 0.85 (SEE=4.64 ml× min-1×kg-1) and 0.85 (SEE=5.02 ml× min-1×kg-1) for women and men, respectively [9].”

Q#2) Line 117: The authors used linear regressions to compare differences in means. The authors did not specify whether they verified the premise of using linear regression (normal distribution, Homogeneity of variances, Homoscedasticity). I therefore suggest that the authors give these details in their paper.

ASN#2) Thanks for the comments. Data was assessed both visually and through K-S test for normal distribution. In addition, Statistical analyses in Methods are now revised extensively as follows;

“All variables were checked for normality, both visually and through the Kolmogorov-Smirnov test, and subjected to log10 transformation, if necessary, prior to statistical analyses. Multiple imputations by chained equation (MICE) was performed to handle the missing data of the study variables identified (<26%) prior to the primary analyses. The number of imputations was determined to be four. Descriptive statistics were presented as means and standard deviations and frequencies and percentages for continuous and categorical variables, respectively. Linear regressions and Chi-square were used to compare differences in means and percentages of measured variables, respectively. The Kaplan-Meier procedure with log-rank tests was used to estimate mortality functions according to baseline eCRF categories. Survival time was measured as the time from the baseline survey to death or the censor point (August 12, 2011). Cox proportional hazards regression model was used to estimate the hazard ratio (HR) and 95% confidence interval (95% CI) of baseline eCRF categories for all-cause mortality as censored observation before and after adjustments for all measured covariates (i.e., socio-demographic variables, health behaviors and conditions, and chronic conditions). All analyses were performed taking into account complex sampling weights, using SPSS-PC 23.0 (SPSS Inc., Chicago, IL, USA).”

Q#3) Lines 134-135: How the authors explain the lack of association between eCRF and cancers in men.

ANS#3) Thanks for the comment. We suspect that lower BMI found in those with cancers might have played a role in overestimating CRF since BMI is one of the regression equation components for the estimation of CRF, weakening the association between eCRF and cancers. In addition, as suggested by reviewer #2 (”I don’t know why cancer outcomes are included in the introduction because they are not really related to the exposure or outcome”), we decided not to report cancers outcomes.

Q#4) I suggest to the authors to enlarge Figure 1

ANS#4) Thanks. Figure 1 is now enlarged.

Q#5) Line 241: I think it's complications instead of complicstions.

ANS#5) Thanks. It is now corrected as complications.

Q#6) Lines 238-244: The authors need to delve deeper into each possible explanation of the correlation between eCRF and all-cause mortality. For example how adopting healthy lifestyles featuring of healthy diets and physical activity enhance cognitive function?

ANS#6) Thanks for the comments. As stated already in the manuscript, several explanations are be given for the association between eCRF and all-cause mortality;

“Several explanations can be given for the inverse association between eCRF and all-cause mortality observed in the current study. The protective effect of eCRF against premature death from all causes might be achieved by adopting healthy lifestyles featuring of healthy diets and physical activity, which, in turn, reduce metabolic complications including elevated blood pressure [30] and fasting glucose [31], and atherogenic lipids [32], improve insulin sensitivity [33], vascular health [34], and immune function [35], control body weight [36], enhance cardiorespiratory function [37] and cognitive function [38]. In a long-term perspective, those health benefits would decrease the risk of being exposed to various chronic diseases, as well as the risk of premature death from all causes and specific causes.

However, we are reluctant to speculate the explanations further due to the cross-sectional nature of the study.

Reviewer 2 Report

IJERPH-494957 presented results from a larger sample of older Koreans. While some parts of this manuscript were interesting, other areas could be improved. I hope the authors consider my feedback for enhancing their manuscript.

MAJOR COMMENTS

Title, Introduction, and Discussion: The results are presented as sex stratified so I think revising these sections to better fit the importance of the research question and implications of the results is necessary. Otherwise, sex stratification for the results seems like an afterthought. Consider major revision in these areas.

Line 68: Excluding participants for missing data or attrition is no longer a statistically sound practice. I recommend the use of multiple imputation for this. Consider performing these analyses, explain the method in the statistical analysis section, and present the results as a sensitivity analysis?

Statistical Analysis: Information for the K-M estimators is not in this section. Necessary details need to be added. Similarly, specific information for each of the Cox models is needed. What were the covariates included in the models? What was the time and entry variable? Was there information for censoring and truncation?

Lines 148-149: Can mortality rate per 1,000 person years be added here?

Results: I would recommend presenting the exposure variable for the Cox models with “Upper” as the reference. During aging, health factors (e.g., muscle strength, CRF) typically decline. Modeling with “Upper” as the reference better depicts the disabling process over time. Edit throughout where appropriate.

Table 3: Why are there 4 models presented. Examining how the covariates attenuated the estimates for the Cox models was not part of the study purpose. Just present the results of the fully-adjusted models?

MINOR COMMENTS

Lines 31-34: I don’t know why cancer outcomes are included in the introduction because they are not really related to the exposure or outcome. Consider revision.

Line 35: You could probably delete, “it is”.

Lines 35-37 could undergo minor revisions for grammar.

Line 41: Should be, “associated with a”. Be sure to double check for grammatical edits throughout.

Line 49: I would recommend softer language here.

Line 70: Descriptive characteristics (57% women) are better for when Table 1 is introduced.

Lines 92-94: This information is better for the statistical analysis section.

Determination of all-cause mortality: Can you provide information for the accuracy of determining mortality in the manuscript (e.g., these sources have demonstrated 99% accuracy)?

Line 114: Please provide manufacturer information for the digital scale.

Covariates: Can the authors include information about how experiences of falling and hospitalization were measured. The same level of detail should be added for all covariates measured.  

Please make sure that all tables and figures stand-alone. For example, the abbreviation ml/min/kg in Table 1 has not been defined in a table note.

Conclusions: Can you add to this section? It is currently a little sparse. For example, are there any practical applications for your findings?

Make any changes to the abstract that align with those made in the text.

Author Response

(Reviewer#2)

Q#1) Title, Introduction, and Discussion: The results are presented as sex stratified so I think revising these sections to better fit the importance of the research question and implications of the results is necessary. Otherwise, sex stratification for the results seems like an afterthought. Consider major revision in these areas.

ANS#1) Thanks for the comments. Title is now changed to Cardiorespiratory fitness without exercise testing can predict all-cause mortality risk in a representative sample of Korean older adults, as suggested by Reviewer #4 too.

In addition, gender difference was not our interest to study. Gender-specific regression models were used to estimate and category CRF levels. So the outcomes are presented separately in Tables 1 and 2. In addition, we found no gender difference in the association between eCRF and all-cause mortality. In this revised manuscript, however, Tables 1-2 are collapsed into Table 1 with the total group being added in another column. HRs of the total group is also added to Table 3 (now renamed as Table 2).  Abstract, Introduction, Results are now revised substantially (refer to Abstract, Introduction, and Results highlighted in Yellow Color). Discussion is revised accordingly.

Q#2) Line 68: Excluding participants for missing data or attrition is no longer a statistically sound practice. I recommend the use of multiple imputation for this. Consider performing these analyses, explain the method in the statistical analysis section, and present the results as a sensitivity analysis?

ANS#2) Thanks for the comments. In response to the comments, we actually performed the primary analyses using multiple imputations for the variables with missing information at baseline using multiple imputations by changed equation (MICE) under the assumption that data were missing at random. So, Statistical Analyses in Methods are extensively revised as follows;

“All variables were checked for normality, both visually and through the Kolmogorov-Smirnov test, and subjected to log10 transformation, if necessary, prior to statistical analyses. Multiple imputations by chained equation (MICE) was performed to handle the missing data of the study variables identified (<26%) prior to the primary analyses. The number of imputations was determined to be four. Descriptive statistics were presented as means and standard deviations and frequencies and percentages for continuous and categorical variables, respectively. Linear regressions and Chi-square were used to compare differences in means and percentages of measured variables, respectively. The Kaplan-Meier procedure with log-rank tests was used to estimate mortality functions according to baseline eCRF categories. Survival time was measured as the time from the baseline survey to death or the censor point (August 12, 2011). Cox proportional hazards regression model was used to estimate the hazard ratio (HR) and 95% confidence interval (95% CI) of baseline eCRF categories for all-cause mortality as censored observation before and after adjustments for all measured covariates (i.e., socio-demographic variables, health behaviors and conditions, and chronic conditions). All analyses were performed taking into account complex sampling weights, using SPSS-PC 23.0 (SPSS Inc., Chicago, IL, USA).”

Q#3) Statistical Analysis: Information for the K-M estimators is not in this section. Necessary details need to be added. Similarly, specific information for each of the Cox models is needed. What were the covariates included in the models? What was the time and entry variable? Was there information for censoring and truncation?

ANS#3) Thanks for the comments. In response to the comments, Statistical Analyses in Methods are extensively revised (please refer to our reply to Q#2).

Q#4) Lines 148-149: Can mortality rate per 1,000 person years be added here?

ANS#4) Mortality rates are now expressed as suggested (i.e., the number of deaths per 1,000 person-years);

In total, the mortality rate per 1,000 person-years was 3.0 for the lower eCRF group, 2.9 for the middle eCRF group, and 1.1 for the upper eCRF group. In men, the mortality rate per 1,000 person-years was 1.2 for the lower eCRF group, 0.6 for the middle eCRF group, and 0.5 for the upper eCRF group. In women, the mortality rate per 1,000 person-years was 0.6 for the lower eCRF group, 0.3 for the middle eCRF group, and 0.2 for the upper eCRF group.

Q#5) Results: I would recommend presenting the exposure variable for the Cox models with “Upper” as the reference. During aging, health factors (e.g., muscle strength, CRF) typically decline. Modeling with “Upper” as the reference better depicts the disabling process over time. Edit throughout where appropriate.

ANS#5) Thanks for the comment. We re-run the Cox regression with upper as reference (refer to Table 2).

Q#6) Table 3: Why are there 4 models presented. Examining how the covariates attenuated the estimates for the Cox models was not part of the study purpose. Just present the results of the fully-adjusted models?

ANS#6) Thanks for the comments. As suggested, unadjusted and fully-adjusted models only are  now presented as the outcomes (refer to Table 2).

Q#7) Lines 31-34: I don’t know why cancer outcomes are included in the introduction because they are not really related to the exposure or outcome. Consider revision.

ANS#7) Thanks. We agreed. Cancer outcomes are now removed.

Q#8) Line 35: You could probably delete, “it is”.

ANS#8) Thanks. “it is” is now removed.

Q#9) Lines 35-37 could undergo minor revisions for grammar.

ANS#9) Thanks. Lines 35-37 are now revised as follows;

While superior to estimated CRF (eCRF), objectively measuring CRF is often limited by the need of specialized equipment and trained personnel and time-consuming [4].

Q#10) Line 41: Should be, “associated with a”. Be sure to double check for grammatical edits throughout.

ANS#10) The sentences are corrected and revised as follows;

Although mostly limited to Western populations, some studies showed that eCRF was inversely associated with mortality from all causes and specific causes. For example, the outcomes of the 1999-2006 National Health and Nutrition Examination Survey [10] and the Aerobics Center Longitudinal Study [11] showed that eCRF was significantly and inversely associated with all-cause and CVD mortality.

Q#11) Line 49: I would recommend softer language here.

ANS#11) Thanks. The sentences are softened as follows;

“Declines in the physical ability to tolerate daily activities may predict mobility problems and mortality from all causes and specific causes, particularly in the sedentary older populations [13, 14]. Thus, maintaining a healthful level of CRF via physical activity is of paramount importance in attenuating morbidity and mortality from various chronic diseases and thereby reducing their societal and economic burden.”

Q#12) Line 70: Descriptive characteristics (57% women) are better for when Table 1 is introduced.

ANS#12) Thanks for the suggestion. However, we believe that the statement is part of study design description and the current position of the statement is appropriate.

Q#13) Lines 92-94: This information is better for the statistical analysis section.

ANS#13) Thanks for the suggestion. Again, the sentences describe the reliability of the regression equation for the estimation of CRF used in the current study. So we believe that the current position of the statements is appropriate.

Q#14) Determination of all-cause mortality: Can you provide information for the accuracy of determining mortality in the manuscript (e.g., these sources have demonstrated 99% accuracy)?

ANS#14) We have no doubt about the accuracy of information because death was identified by death certificate filed upon at the registry office of the municipality of residence.

Q#15) Line 114: Please provide manufacturer information for the digital scale.

ANS#15) Thanks. Height and weight were simultaneously measured with a digital scale. The sentence is now revised as follow;

“Height and body weight were measured using a portable digital scale (JENIXÒ, Dong  Shan Jenix Co., TLD., Seoul, Korea), after removing the shoes and wearing only light clothing. BMI was calculated by dividing body weight by height squared (kg/m2).

Q#16) Covariates: Can the authors include information about how experiences of falling and hospitalization were measured. The same level of detail should be added for all covariates measured.

ANS#16 ) Thanks for the comments. Detailed procedures are described in our previous studies, which are now cited as additional references (Ann Hum Biol 2018, 45, 337-345 and Geriatr Gerontol Int 2018, 186, 950-956.).

Q#17) Please make sure that all tables and figures stand-alone. For example, the abbreviation ml/min/kg in Table 1 has not been defined in a table note.

ANS#17) Thanks. In response to the comment, ml×kg-1×min-1 is now defined as milliliter (ml) of oxygen consumption per minute (min) per body weight (kg) in Table 1.

Q#18) Conclusions: Can you add to this section? It is currently a little sparse. For example, are there any practical applications for your findings?

Q#18) Thanks. 5. Conclusions are now revised as follows;

Low CRF is well known as an independent indicator of premature death from all and specific causes. However, objectively measuring CRF is usually limited due to mobility issues, chronic conditions, and other health complications in elderly persons. In this context, therefore, we investigated the role of CRF without exercise testing in the prediction of all-cause mortality in a representative sample of Korean older adults and found that a higher eCRF was significantly associated with a lower risk of all-cause mortality, implying a predictive value of eCRF for all-cause mortality in Korean older adults.

Q#19) Make any changes to the abstract that align with those made in the text.

ANS#19) Thanks. Abstract is completely changed to incorporate changes made in the text;

“This study examined the association between cardiorespiratory fitness (CRF) without exercise testing and all-cause mortality in Korean older adults. The present study was carried out using data from the 2008 and 2011 Living Profiles of Older People Survey. A total of 14,122 participants aged 60 years and older (57% women) completed the 2008 baseline and 2011 followup assessments (i.e., socioeconomic status, health behaviors and conditions, and prevalence of chronic diseases), and they were included for the final analyses. CRF was estimated (eCRF) with sex-specific algorithms and classified as lower (lowest 25%), middle (middle 50%), and upper (highest 25%). Cox proportional hazards regression was used to calculate hazard ratios (HRs) and 95% confidence intervals (CIs) across eCRF categories. In total, multivariable-adjusted HRs and 95% CIs were 1 for the upper eCRF group (referent), 1.059 (0.814~1.378) for the middle eCRF group, and 1.714 (1.304~2.253) for the lower eCRF group. In men, multivariable-adjusted HRs and 95% CIs were 1 for the upper eCRF group (referent), 1.011 (0.716~1.427) for the middle eCRF group, and 1.566 (1.098~2.234) for the lower eCRF group. In women, multivariable-adjusted HRs and 95% CIs were 1 for the upper eCRF group (referent), 1.064 (0.707~1.602) for the middle eCRF group, and 1.599 (1.032~2.478) for the lower eCRF group. Together, the current findings suggest that eCRF may have an independent predictor of all-cause mortality, underscoring the importance of promoting physical activity to maintain a healthful level of CRF in Korean geriatric population.

Reviewer 3 Report

The investigators conducted a population-based prospective study to show that a higher eCRF was significantly associated with a lower risk of all-cause mortality in Korean geriatric population. I have some comments.

1.     Please provide further supplementary data regarding the most common three causes of death among all study participants with different eCRF.

2.     The follow-up duration was probably too short to avoid the issue of reverse causation. I suggest a sensitivity analysis during 12 and 24 months of follow-up respectively for Table 3 (Model 3) to confirm risks of all-cause mortality according to eCRF category.

3.     Line 37-39: do citation.

4.     Line 41: should be revised as “… associated ‘with’ a lower risk…”.

Author Response

(Reviewer #3)

Q#1) Please provide further supplementary data regarding the most common three causes of death among all study participants with different eCRF.

ANS#1) Thanks for the suggestion. Unfortunately, however, data on specific causes of death were not available in the LPOPS datasheet.

Q#2) The follow-up duration was probably too short to avoid the issue of reverse causation. In suggest a sensitivity analysis during 12 and 24 months of follow-up respectively for Table 1 (Model 3) to confirm risks of all-cause mortality according to eCRF category.

ANS#2) Agreed. In response to the concern, we conducted a sensitivity analysis by excluding all events that occurred the first one year of follow-up. The outcomes were similar to those of the primary analyses and added to Table 2.

Table 2   Risks of all-cause mortality according to eCRF category (HR and 95% CI)

3-year mortality risk

2-year mortality risk

Model 1

Model 2

Model 1

Model 2

Total (N=14,122)

Upper

1 (reference)

1 (reference)

1 (reference)

1 (reference)

Middle

1.313 (1.048-1.645)

1.059 (0.814-1.378)

1.445 (1.106-1.889)

1.189 (0.868-1.628)

Lower

2.648 (2.121-3.307)

1.714 (1.304-2.253)

2.810 (2.156-3.663)

1.883 (1.358-2.612)

<0.001

<0.001

<0.001

<0.001

Men (N=6,105)

Upper

1 (reference)

1 (reference)

1 (reference)

1 (reference)

Middle

1.253 (0.936-1.676)

1.011 (0.716-1.427)

1.366 (0.969-1.926)

1.084 (0.724-1.623)

Lower

2.227 (1.673-2.967)

1.566 (1.098-2.234)

2.252 (1.601-3.168)

1.653 (1.090-2.508)

P for linear trends

<0.001

<0.001

<0.001

<0.001

Women (N=8,017)

Upper

1 (reference)

1 (reference)

1 (reference)

1 (reference)

Middle

1.393 (0.976-1.989)

1.064 (0.707-1.602)

1.558 (1.106-2.390)

1.274 (0.769-2.112)

Lower

3.056 (2.151-4.343)

1.599 (1.032-2.478)

3.479 (2.281-5.306)

1.865 (1.089-3.195)

P for linear trends

<0.001

<0.001

<0.001

<0.001

eCRF:   estimated cardiorespiratory fitness; HR: hazard ratio; CI=confidence   interval. Model 1: Unadjusted; Model 2: Adjusted for socioeconomic   status (i.e., education and income), socioeconomic status and health   behaviors (i.e., solitary, nutritional risk, smoking, alcohol consumption)   and conditions (i.e., frailty, impaired ADL, hospitalization, falling,   impaired cognition, and depressive symptoms), and chronic diseases (i.e., hypertension, diabetes,   CAD, stroke, lung disease, and cancer).

 In addition, Description of Table 2 is now revised as follows;

“Table 2 represents the associations between eCRF and all-cause mortality. In total, trends test showed significant and linear trends in both 3- and 2-year all-cause mortality across incremental eCRF categories (p<0.001 and p<0.001, respectively); considering the upper eCRF group as reference (HR=1), the lower and middle CRF groups had a higher 3-year all-cause mortality risk (HR=2.648 and 95% CI=2.121~3.307; HR=1.313 and 95% CI=1.048~1.645, respectively) as well as a 2-year all-cause mortality risk (HR=2.810 and 95% CI=2.156~3.663; HR=1.445 and 95% CI=1.106~1.889, respectively). The linear trend for crude HRs remained statistically significant even after adjustments for socioeconomic status, health behaviors and conditions, and prevalence of chronic diseases. In men, trends test showed significant and linear trends in both 3- and 2-year all-cause mortality across incremental eCRF categories (p<0.001 and p<0.001, respectively); considering the upper eCRF group as reference (HR=1), the lower and middle CRF groups had a higher 3-year all-cause mortality risk (HR=2.227 and 95% CI=1.673~2.967; HR=1.253 and 95% CI=0.936~1.676, respectively) as well as a 2-year all-cause mortality risk (HR=2.252 and 95% CI=1.601~3.168; HR=1.366 and 95% CI=0.969~1.926, respectively). The liner trend for crude HRs remained statistically significant even after adjustments for all the covariates. In women, trends test showed significant and linear trends in 3- and 2-year all-cause mortalities across incremental eCRF categories (p<0.001 and p<0.001, respectively); considering the upper eCRF group as reference (HR=1), the lower and middle eCRF groups had a higher 3-year all-cause mortality risk (HR=3.056 and 95% CI=2.151~4.343; HR=1.393 and 95% CI=0.976~1.989) as well as a 2-year all-cause mortality risk (HR=3.479 and 95% CI=2.281~5.306; HR=1.558 and 95% CI=1.106~2.390, respectively). The linear trend for crude HRs remained statistically significant even after adjustments for all the covariates.

Q#3) Line 37-39: do citation.

ANS#3). Thanks. References are now cited. “However, the findings of recent studies showed that CRF could be estimated from easily obtained health indicators with an acceptable accuracy [9].

Q#4) Line 41: should be revised as”…. Associated with a lower risk…” The investigators conducted a population-based prospectively study to

ANS#4) Thanks. It is now corrected as suggested;

Although mostly limited to Western populations, some studies showed that eCRF was inversely associated with mortality from all causes and specific causes. For example, the outcomes of the 1999-2006 National Health and Nutrition Examination Survey [10] and the Aerobics Center Longitudinal Study [11] showed that eCRF was significantly and inversely associated with all-cause and CVD mortality.

Reviewer 4 Report

Dear authors,

It is well-designed and interesting study.

I would suggest reconsider the main predictors’ entitlement from “non-exercise CRF” into “CRF without exercise testing” like it was provided in the reference cited. In current situation after reading the manuscript title it could be interpreted that CRF assessment used was based on other factors (eg. genetics) but not on physical activity or exercise. In your case CRF was based only on self-reported physical activity so it should be corrected thorough in the whole manuscript.

What about cardiovascular mortality and CRF? Could it be added to the results?

Discussion section must incorporate interpretation of CRF without exercise testing prognostic value on mortality risk based on research of other authors. Introduction section should be supplemented by the mechanisms explaining the relations between self-reported physical activity and CRF in elderly population.

Prospective study design allows to change “associations” into “mortality risk” in the study title, eg. “Cardiorespiratory fitness without exercise testing can predict all-cause mortality risk in a representative sample of Korean older adults”.

Please reconsider this suggestions or provide alternative more clear interpretation of the major study predictor.

Author Response

(Reviewer #4)

Q#1) I would suggest reconsider the main predictors’ entitlement from “non-exercise CRF into CRF without exercise testing” like it was provided in the reference cited. In current situation after reading the manuscript title it could be interpreted that CRF assessment used was based on other factors (e.g., genetics) but not on physical activity or exercise. In your case CRF was based on only self-reported physical activity so it should be corrected through in the whole manuscript.

ANS#1) Thanks for the thoughtful comments. We agreed. Non-exercise CRF is changed as CRF without exercise testing in Tittle and throughout the manuscript.

Q#2) What about cardiovascular mortality and CRF? Could it be added to the results?

ANS#2) Thanks for the suggestion. Unfortunately, however, data on specific causes of death were not available in the LPOPS datasheet.

Q#3) Discussion section must incorporate interpretation of CRF without exercise testing prognostic value on mortality risk based on research of other authors. Introduction section should be supplemented by the mechanisms explaining the relations between self-reported physical activity and CRF in elderly population.

ANS#3) Thanks for the comments. Introduction are substantially revised (refer to Introduction highlighted in yellow color).

“Cardiorespiratory fitness (CRF) is a key component of health-related physical fitness and reflects the ability of the respiratory and circulatory systems to deliver oxygen to working skeletal muscles during exercise. Maintenance and enhancement of CRF influence several biologic mechanisms in such way that the risk of various chronic diseases may be reduced significantly [1, 2]. In addition, CRF was found to alter the relationship between adiposity and its health outcomes, as illustrated in so called the obesity paradox [3]. In this context, CRF is also known as a strong predictor of all-cause and cardiovascular disease mortality [4], cancer mortality [5] as well as incidence of various chronic diseases [6], supporting its diagnostic and prognostic values in the prediction of premature death from all causes and specific causes. While superior to estimated CRF (eCRF), objectively measuring CRF is often limited by the need of specialized equipment and trained personnel and time-consuming [7] as well as mobility problems and chronic conditions in elderly persons [8]. However, the findings of recent studies showed that CRF could be estimated from easily obtained health indicators with an acceptable accuracy [9]. Thus, CRF without exercise testing may provide an alternative approach for routine clinical screening and health promotion with the purpose to identify individuals with poor CRF who are at increased health risk [10, 11]. Although mostly limited to Western populations, some studies showed that eCRF was inversely associated with mortality from all causes and specific causes. For example, the outcomes of the 1999-2006 National Health and Nutrition Examination Survey [10] and the Aerobics Center Longitudinal Study [11] showed that eCRF was significantly and inversely associated with all-cause and CVD mortality. In a Spanish cohort of men and women aged 60 years and older, however, Martinez-Gomez et al. [12] found that eCRF was inversely associated with all-cause mortality in women but not in men, questioning the value of non-exercise eCRF as a diagnostic and/or prognostic tool to predict mortality in elderly adults. Declines in the physical ability to tolerate daily activities may predict mobility problems and mortality from all causes and specific causes, particularly in the sedentary older populations [13, 14]. Thus, maintaining a healthful level of CRF via physical activity is of paramount importance in attenuating morbidity and mortality from various chronic diseases and thereby reducing their societal and economic burden. Yet, little information is available regarding the clinical values of eCRF in predicting mortality of Asian populations, including Koreans. Therefore, this study examined the association between eCRF and all-cause mortality during a 3-year follow-up in a representative sample of Korean adults aged 60 years and older.”

Q#4) Prospective study design allows to change associations into mortality risk in the study title, eg. Cardiorespiratory fitness without exercise testing can predict all-cause mortality risk in a representative sample of Korean older adults. Please reconsider this suggestions or provide alternative more clear interpretation of the major study predictor.

ANS#4) Thanks for the comments/suggestions. We agreed. Title is changed as suggested; “Cardiorespiratory fitness without exercise testing can predict all-cause mortality risk in a representative sample of Korean older adults

Round 2

Reviewer 1 Report

Good work

Author Response

Thanks for your positive comment.

Reviewer 2 Report

Line 36: Please delete, “cancer mortality [5]” and revise appropriately here. Including comments about cancer in the Introduction contradicts the edits you proposed making in the previous version of the manuscript.

Line 82: CRF should be spelt out because it is a sub-header.

Lines 85-86: PAR and BMI have not yet been defined (e.g., body mass index (BMI)). Make sure that all abbreviations are defined first throughout the manuscript.

Lines 159-160: Please include the specific statistic in parentheses for “No association between eCRF and smoking was found.”

Results: Similar to my previous comment, be sure to add specific statistics in the Results section where appropriate. For example, in lines 170-171 the authors suggest, “an association between eCRF and alcohol consumption was found in men only but not in women.” This statement should be accompanied with specific results from the analyses. Please revise in this section accordingly.

Can you make it clearer where in the results section the sensitivity analyses results are presented?

Author Response

In Response to the Comments/Suggestions by Reviewer #2

Q1) Line 36: Please delete, “cancer mortality [5]” and revise appropriately here. Including comments about cancer in the Introduction contradicts the edits you proposed making in the previous version of the manuscript.
ANS1) Thanks for the comment. We deleted “caner mortality [5]” and the number and citation order of references are appropriately revised.

Q2) Line 82: CRF should be spelt out because it is a sub-header.
ANS2) Thanks. CRF is spelled out as cardiorespiratory fitness in the sub-header.
Q30 Lines 85-86: PAR and BMI have not yet been defined (e.g., body mass index (BMI)). Make sure that all abbreviations are defined first throughout the manuscript.
ANS3) Thanks. PAR and BMI are spelled out as physical activity rating and body mass index.
Q4) Lines 159-160: Please include the specific statistic in parentheses for “No association between eCRF and smoking was found.”
ANS4) Linear trend analysis showed that eCRF was not signiticantly associated with smoking (p=0.489).
Q5) Results: Similar to my previous comment, be sure to add specific statistics in the Results section where appropriate. For example, in lines 170-171 the authors suggest, “an association between eCRF and alcohol consumption was found in men only but not in women.” This statement should be accompanied with specific results from the analyses. Please revise in this section accordingly.
ANS5) Gender-specific linear trend analyses showed that eCRF was significantly associated with alcohol consumption in men (p<0.001) but not in women (p=0.133).
Q6) Can you make it clearer where in the results section the sensitivity analyses results are presented?
ANS6) Thanks. In reponse to the commentm, the followiing statement is now added to Statistical analyses section; “The outcomes of the sensity analyses are presented in Figure 1 and Table 2.”

Reviewer 3 Report

Much better. Congratulations.

Author Response

Thanks for your positive comment.

Reviewer 4 Report

Thank you for the interesting research. Awaiting it in press to be cited.

Author Response

Thanks for your positive comment.